# Using Survey Data to Estimate the Impact of the Omicron Variant on Vaccine Efficacy against COVID-19 Infection

### Jesús Rufino
IMDEA Networks Institute
Madrid, Spain

### Carlos Baquero
U. Porto & INESC TEC
Porto, Portugal

### Davide Frey
Univ Rennes, IRISA, CNRS, Inria
Rennes, France

### Christin A. Glorioso
Academics for the Future of Science,
Inc. & U. of California San Francisco
San Francisco, USA

### Antonio Ortega
U. Southern California
Los Angeles, USA

### Nina Reščič
Jožef Stefan Institute, Department of
Intelligent Systems
Ljubljana, Slovenia

### Julian Charles Roberts
Skyhaven Media
Liverpool, UK

### Rosa E. Lillo
U. Carlos III de Madrid
Getafe, Spain

### Raquel Menezes
Centre of Mathematics of U. Minho
Guimarães, Portugal

### Jaya Prakash Champati
IMDEA Networks Institute
Madrid, Spain

### Antonio Fernández Anta
IMDEA Networks Institute
Madrid, Spain

## ABSTRACT

Data collected in the Global COVID-19 Trends and Impact Surveys (UMD Global CTIS), and data on variants sequencing from GISAID, are used to evaluate the impact of the Omicron variant (in South Africa and other countries) on the prevalence of COVID-19 among unvaccinated and vaccinated population, in general and discriminating by the number of doses. In South Africa, we observe that the prevalence of COVID-19 in December (with strong presence of Omicron) among the unvaccinated population is comparable to the prevalence during the previous wave (in August-September), in which Delta was the variant with the largest presence. However, among vaccinated, the prevalence of COVID-19 in December is much higher than in the previous wave. In fact, a significant reduction of the vaccine efficacy is observed from August-September to December. For instance, the efficacy drops from 0.81 to 0.30 for those vaccinated with 2 doses, and from 0.51 to 0.09 for those vaccinated with one dose. The study is then extended to other countries in which Omicron has been detected, comparing the situation in October (before Omicron) with that of December. While the reduction measured is smaller than in South Africa, we still found, for instance, an average drop in vaccine efficacy from 0.53 to 0.45 among those vaccinated with two doses. Moreover, we found a significant negative (Pearson) correlation of around −0.6 between the measured prevalence of Omicron and the vaccine efficacy.

**ACM Reference Format:**
Jesús Rufino, Carlos Baquero, Davide Frey, Christin A. Glorioso, Antonio Ortega, Nina Reščič, Julian Charles Roberts, Rosa E. Lillo, Raquel Menezes, Jaya Prakash Champati, and Antonio Fernández Anta. 2022. Using Survey Data to Estimate the Impact of the Omicron Variant on Vaccine Efficacy against COVID-19 Infection . In *epiDAMIK 2022: 5th epiDAMIK ACM SIGKDD International Workshop on Epidemiology meets Data Mining and Knowledge Discovery, August 15, 2022, Washington, DC, USA.* ACM, New York, NY, USA, 14 pages.

## 1 INTRODUCTION

The Omicron variant B.1.1.529 of SARS-CoV-2 has seen an impressive increase by January 2022 (when this study was completed), since its initial classification in November 2021 [21]. In South Africa it appears to have out-competed the Delta variant [10] and has rapidly spread into Europe and other regions. Preliminary observations also indicate that it might spread faster and might have higher immune evasiveness than previous variants [12]. While vaccination still provides a level of protection against a serious disease [25], recent results [14, 18, 20, 23] point towards a reduced level of protection against infection, especially from 15 weeks post the second dose [3], and it is likely that the number of breakthrough infections (i.e., infections among vaccinated people) will rise with the spread of Omicron. It is also possible that the rapid spread of Omicron is not only a consequence of high transmissibility but also of immune evasiveness [18]. Some of the preliminary models [27] showed that high transmissibility in combination with high immune evasiveness could lead to a concerning health system overload [17].

Since the Spring of 2020, the University of Maryland in collaboration with Facebook has collected extensive survey data on self-reported symptoms, infection, testing, behavior and, more recently, vaccination status (UMD Global CTIS) [5, 30]. In mid December 2021, researchers used data from this survey concerning the Gauteng province in South Africa to define different combinations of symptoms that are associated with COVID-19 infection, and combined those with self-reported vaccination status to compare vaccine efficacy changes from a Delta dominant period to the current Omicron dominant period [31]. Their findings showed a

measurable drop of efficacy towards infection for those vaccinated with two doses.

In this study we use self-reported confirmation of COVID-19 infection, from a subset of the UMD Global CTIS survey responses, to derive an improved proxy for COVID-19 active cases (using a Random Forest classifier) that tracks more closely the evolution of confirmed cases. We use this improved proxy for analysing prevalence and vaccine efficacy changes in South Africa as a whole, and in the Gauteng province, among those unvaccinated, partially vaccinated, and fully vaccinated. We also compute results in other countries that are currently experiencing a rise of Omicron B.1.1.529 cases, which show a significant negative correlation between the prevalence of Omicron and the vaccine efficacy.

## 2 METHODS

### 2.1 Self-reported Survey Data

We use the responses to the UMD Global CTIS, which collects more than 100,000 responses daily across the world (except in the US, where the survey is run by CMU [26]). We have access to the responses collected by agreement with UMD and Facebook (see Appendix A). All the participants in the CTIS have declared to be at least 18 years of age. The first step is removing abnormal responses, as proposed in Alvarez et al. [2]: We remove responses that declare to have all symptoms or that declare unusual values (greater than 100) in the quantitative questions of the survey (e.g., days of symptom duration, number of symptomatic contacts, number of people staying at the same place, etc.). In order to classify the responses as positive or negative, several criteria have been proposed in the literature. In particular, we consider the following symptom-based COVID-like illness classifiers (see Appendix C for the list of symptoms collected in the survey):

- UMD CLI [2, 8]: A response is considered to be positive if it declares fever (symptom B1_1), along with cough (symptom B1_2), or shortness of breath / difficulty breathing (symptom B1_3). Otherwise, it is negative.
- Stringent CLI [31]: A response is positive if it declares anosmia (symptom B1_10), combined with fever (B1_1), muscle pain (B1_6), or cough (B1_2). Otherwise, it is negative.
- Classic CLI [31]: A response is positive if it declares cough (B1_2), combined with fever (B1_1), muscle pain (B1_6), or anosmia (B1_10). Otherwise, it is negative.
- Broad CLI [31]: A response is positive if it declares muscle pain (B1_6), combined with fever (B1_1), cough (B1_2), or anosmia (B1_10). Otherwise, it is negative.

### 2.2 Machine Learning Classifier: Random Forest

The above methods for classifying cases as positive or negative have two main limitations. First, they do not take into account diagnostic uncertainty, e.g., the same set of symptoms might be associated with some other condition. Second, these criteria are not adaptive to possible changes in the symptoms experienced as conditions change, e.g., as vaccination rates increase or new virus variants emerge. Thus, in this work, we introduce a new machine-learning-based classifier: Random Forest[1].

*2.2.1 Ground-truth Set.* After curating the responses, the next task we face is determining whether they correspond to active cases of COVID-19. This is somewhat direct for the subset of responses that respond affirmatively to the survey question "B7: Have you been tested for COVID-19 in the past 14 days?" and then respond positively or negatively to the survey question "B8a: Did your most recent test find that you had COVID-19?" [29]. For this work, we assume that a participant responding affirmatively to both questions is an active case of COVID-19 (i.e., it is a *positive* case). Similarly, a participant responding affirmatively to Question B7 and negatively to Question B8a is assumed not infected with COVID-19 (i.e., *negative*). This set of classified responses constitute a *ground-truth set*, for which infection status (positive or negative) is available. Observe that some positive cases in the ground truth are asymptomatic.

Unfortunately, this ground-truth set cannot be used directly to estimate the prevalence of COVID-19 in the overall population, because the set is usually very small and is not produced via uniform random sampling: People who have reason to believe they may be infected are more likely to be tested and therefore the ratio of positives among those tested in the latest 14 days (i.e., the *testing positive rate*, abbreviated TPR) is higher than the actual prevalence.

*2.2.2 Creating the Machine Learning Classifier: Random Forest.* Each response to the survey includes a large number of questions. Obviously, not all participants answer all questions, but a large fraction responds to the most important questions. For instance, in December 2021, out of 19, 740 responses from South Africa, 19, 014 reported whether they had had COVID-19 or not. For training and inference of the Random Forest classifier, we use only questions with answers holding discrete values. From these we remove questions B7 and B8a, which are only used to create the ground-truth set, as well as related questions, such as "B0: As far as you know, have you ever had coronavirus (COVID-19)?" and "B15: Do any of the following reasons describe why you were tested for COVID-19 in the past 14 days?". Finally, we do not use the questions related to vaccination, since we do not want them to influence the classification. The set of questions used can be found in Appendix D, and includes almost 100 different questions (not only symptoms). The answers to this set of questions are "dummified" before they are used, i.e., a question with $k$ possible answers (one of which can be NA/NaN) is replaced by $k$ binary attributes. The Random Forest model is generated with the randomForest function in R. No hyperparameter tuning is done, and the standard options of the function are used, with the exception of limiting the model to 100 trees to reduce the training time.

Observe that the set of questions includes all symptoms, but also has many more questions, including behavioral or demographic aspects. Since the ground truth contains asymptomatic positive cases, the Random Forest classifier also identifies positive asymptomatic cases. Moreover, the Random Forest classifier can give different weights to different symptoms, while previously proposed symptom based criteria are based on determining only whether a symptom is

---

[1]We have experimented with other ML algorithms, like XGBoost, and the results are almost the same as those obtained with Random Forest.

| Country | Quarter | Classifier | Accuracy | Sensitivity | Specificity | F-score |
|---------|---------|------------|----------|-------------|-------------|---------|
| Argentina | 2021-Q3 | Random Forest | **0.85** | 0.80 | **0.86** | **0.61** |
| | | UMD CLI | 0.78 | 0.74 | 0.79 | 0.25 |
| | | Stringent CLI | 0.82 | **0.85** | 0.82 | 0.44 |
| | | Classic CLI | 0.81 | 0.67 | 0.83 | 0.48 |
| | | Broad CLI | 0.80 | 0.64 | 0.82 | 0.45 |
| Japan | 2021-Q3 | Random Forest | **0.95** | **0.81** | **0.96** | **0.51** |
| | | UMD CLI | 0.94 | 0.58 | 0.95 | 0.36 |
| | | Stringent CLI | **0.95** | 0.77 | 0.95 | 0.39 |
| | | Classic CLI | 0.93 | 0.44 | **0.96** | 0.42 |
| | | Broad CLI | 0.91 | 0.29 | 0.95 | 0.29 |
| South Africa | 2021-Q3 | Random Forest | **0.83** | 0.81 | **0.83** | **0.71** |
| | | UMD CLI | 0.71 | 0.70 | 0.72 | 0.34 |
| | | Stringent CLI | 0.79 | **0.87** | 0.77 | 0.57 |
| | | Classic CLI | 0.77 | 0.71 | 0.80 | 0.61 |
| | | Broad CLI | 0.76 | 0.70 | 0.78 | 0.57 |
| Argentina | 2021-Q4 | Random Forest | **0.90** | **0.71** | **0.91** | **0.51** |
| | | UMD CLI | 0.88 | 0.63 | 0.89 | 0.35 |
| | | Stringent CLI | 0.88 | 0.70 | 0.89 | 0.37 |
| | | Classic CLI | 0.86 | 0.48 | **0.91** | 0.44 |
| | | Broad CLI | 0.86 | 0.47 | 0.90 | 0.42 |
| Japan | 2021-Q4 | Random Forest | **0.97** | **0.69** | **0.97** | **0.31** |
| | | UMD CLI | 0.96 | 0.26 | **0.97** | 0.20 |
| | | Stringent CLI | **0.97** | 0.59 | **0.97** | 0.30 |
| | | Classic CLI | 0.94 | 0.18 | **0.97** | 0.22 |
| | | Broad CLI | 0.93 | 0.11 | **0.97** | 0.14 |
| South Africa | 2021-Q4 | Random Forest | **0.83** | 0.69 | **0.85** | **0.55** |
| | | UMD CLI | 0.79 | 0.63 | 0.81 | 0.35 |
| | | Stringent CLI | 0.80 | **0.74** | 0.80 | 0.32 |
| | | Classic CLI | 0.80 | 0.58 | 0.84 | 0.48 |
| | | Broad CLI | 0.80 | 0.58 | 0.84 | 0.47 |

**Table 1: Performance for three different countries in two different 3-month periods (2021-Q3: July-September 2021 and 2021-Q4: October-December 2021) of the different classifiers in the ground-truth set, when randomly divided into training (70%) and testing (30%) subsets. The highest value for each country, period, and metric is shown in bold.**

present or not. Thus, overall the Random Forest classifier is much more versatile than the symptom-based criteria described in the previous section. Additionally, there are other aspects that make the Random Forest classifier(s) more adaptive: (1) We create different models for different countries. It is expected that different countries will have local characteristics, thus training a different classifier for each country can capture them. (2) We create not one but several models per country: one for each 3-month period. This allows the model to capture and adapt to aspects that change over time, like the level of vaccination, the surge of new variants, or the stringency of measures imposed.

*2.2.3 Evaluating the Classifiers.* In order to verify whether the Random Forest classifier provides better proxy estimates than the symptoms-based classifiers, we selected a set of countries and tested the performance of each classifier in the last two quarters of 2021. To this end, we randomly divided the ground-truth set into a training and a testing set, with 70% and 30% of the responses of the ground-truth set in each subset, respectively. Table 1 shows the results for three countries that have detected Omicron in December

for the periods of July-September 2021 (2021-Q3) and of October-December 2021 (2021-Q4). The classification performance metrics used are *accuracy* (ratio of cases correctly classified over the size of the test set), *sensitivity / recall* (ratio of cases correctly classified as positive over the number of positive cases), *specificity* (ratio of cases correctly classified as negative over the number of negative cases), and *F-score* (harmonic mean of precision and recall, where the precision is the ratio of cases correctly classified as positive over the number of all cases classified as positive). As can be seen in Table 1, Random Forest almost always shows the highest performance (marked in bold) among the classification methods used.

As another test, we then selected a set of countries that includes South Africa, along with the 20 countries that have the largest number of available responses in the UMD Global CTIS dataset. For each of these countries, the first two columns of Table 2 show the official Test Positivity Rates obtained via *Our World In Data* [22, 24] (OWID TPR) and the corresponding survey-based estimate from the UMD Global CTIS dataset (CTIS TPR). The remaining columns show the Pearson correlation coefficient between the time series of Confirmed active cases (computed based on data from Johns

| | | | Pearson correlation with Confirmed | | | | |
|---|---|---|---|---|---|---|---|
| Country | OWID TPR | CTIS TPR | Random Forest | UMD CLI | Stringent CLI | Classic CLI | Broad CLI |
| Argentina | **0.09** | 0.17 | **0.95** | **0.97** | **0.96** | **0.92** | **0.91** |
| Australia | **0.01** | **0.02** | **0.93** | 0.46 | 0.31 | -0.10 | 0.03 |
| Brazil | – | 0.19 | **0.98** | 0.03 | 0.82 | 0.36 | 0.46 |
| Canada | **0.03** | **0.04** | **0.94** | 0.85 | 0.66 | 0.73 | 0.71 |
| France | **0.03** | **0.05** | **0.92** | 0.69 | 0.80 | 0.57 | 0.61 |
| Germany | **0.09** | **0.01** | **0.96** | 0.88 | **0.91** | 0.82 | 0.81 |
| Hungary | **0.08** | 0.16 | **0.93** | 0.85 | **0.95** | 0.82 | 0.79 |
| India | **0.02** | 0.16 | 0.31 | -0.38 | -0.31 | -0.71 | -0.37 |
| Italy | **0.02** | **0.03** | **0.98** | 0.86 | 0.85 | 0.71 | 0.72 |
| Japan | **0.05** | **0.04** | **0.93** | **0.90** | 0.84 | -0.17 | 0.67 |
| Mexico | 0.27 | 0.22 | **0.97** | **0.99** | **0.98** | **0.95** | **0.98** |
| Poland | **0.08** | 0.16 | **0.96** | 0.82 | **0.97** | 0.80 | 0.80 |
| Romania | **0.07** | **0.09** | **0.94** | **0.96** | **0.98** | **0.96** | **0.95** |
| Russia | **0.05** | 0.14 | 0.38 | 0.34 | 0.37 | 0.41 | 0.33 |
| South Africa | 0.16 | 0.24 | **0.93** | **0.92** | 0.84 | **0.97** | **0.98** |
| Spain | **0.07** | **0.09** | **0.93** | 0.82 | 0.79 | 0.48 | 0.52 |
| Sweden | **0.06** | **0.05** | **0.91** | 0.83 | 0.74 | 0.71 | 0.67 |
| Thailand | 0.20 | **0.07** | 0.85 | 0.83 | **0.92** | 0.84 | 0.77 |
| Ukraine | 0.20 | 0.16 | **0.97** | 0.87 | **0.95** | **0.91** | 0.89 |
| United Kingdom | **0.04** | **0.06** | 0.84 | 0.70 | 0.52 | 0.59 | 0.60 |
| Vietnam | **0.06** | **0.02** | 0.83 | 0.79 | 0.79 | 0.74 | 0.78 |

**Table 2: Test-positivity rate (TPR) obtained from OWID and extracted from the UMD Global CTIS data for the 20 countries with largest survey data and South Africa. Values of at most 0.1 are shown in bold. The rest of columns show the Pearson correlation coefficient of each different proxy with the Confirmed time series. Correlation values of at least 0.9 are shown in bold. The time period used is Jun 18th, 2021 to Dec 31st, 2021. The estimates have been smoothed with a rolling average of 14 days.**

Hopkins University [11] as described by Alvarez et al. [2]) and that of each of the candidate proxies in the period June 18th, 2021 (start of the first period considered in [31]), to December 31st, 2021. All time series have one value per day, which is the average of the latest 14 days.

We can make two observations from Table 2. First, among all candidate proxies considered, Random Forest achieves at least 0.9 correlation for the largest number of countries. Second, 17 out of the 21 countries exhibit low TPR ($\leq 0.1$) values in at least one of the first two columns (either official or survey-based TPR), and 11 out of the 21 exhibit low values in both columns, with 7 having values no higher than 0.05 (the WHO considers countries to have the epidemic under control when their TPR is below 0.05 [32]). This suggests that such countries keep the case count under control and report more accurate official data on confirmed cases. We can thus interpret the higher correlation between the Random Forest proxy and the Confirmed time series for the countries with low TPR as a sign that this proxy constitutes the most promising option among the five proxies considered, and thus will also be more accurate for countries for which the official data will be less reliable.

## 2.3 Prevalence and Efficacy Estimation

The prevalence of COVID-19 estimated by a given classifier is the ratio between the number of positive cases over the total number of responses. Then, we consider four subsets of responses:

- Unvaccinated: Participants that respond negatively to the question "V1: Have you had a COVID-19 vaccination?"
- Vaccinated: Participants that respond positively to Question V1.
- Vaccinated with 1 dose: Participants that respond positively to Question V1 and declare having received 1 dose in Question "V2: How many COVID-19 vaccinations have you received?"
- Vaccinated with 2 doses: Participants that respond positively to Question V1 and declare having received 2 doses in Question V2.

Unfortunately, from the questions in the UMD Global CTIS it is not possible to know whether those with one dose are fully vaccinated, i.e., they have received a one-dose vaccine, or they simply received only the first dose of a two-dose vaccination. Similarly, it is not possible to know whether a survey respondent received a booster shot. For each of the above subsets, the prevalence of COVID-19 is computed as the fraction of responses classified as positive among the responses that report a given vaccination status. For each proxy we also estimate the *vaccine efficacy* ($V_E$) against illness as in [31], based on the estimates of prevalence among unvaccinated ($P_U$) and vaccinated ($P_V$):

$$V_E = 1 - P_V/P_U.$$

The confidence intervals of this metric are obtained using the Katz-log Method [1]. Since we have three subsets of vaccinated participants, we compute the vaccine efficacy for the subsets Vaccinated, Vaccinated with 1 dose, and Vaccinated with 2 doses.

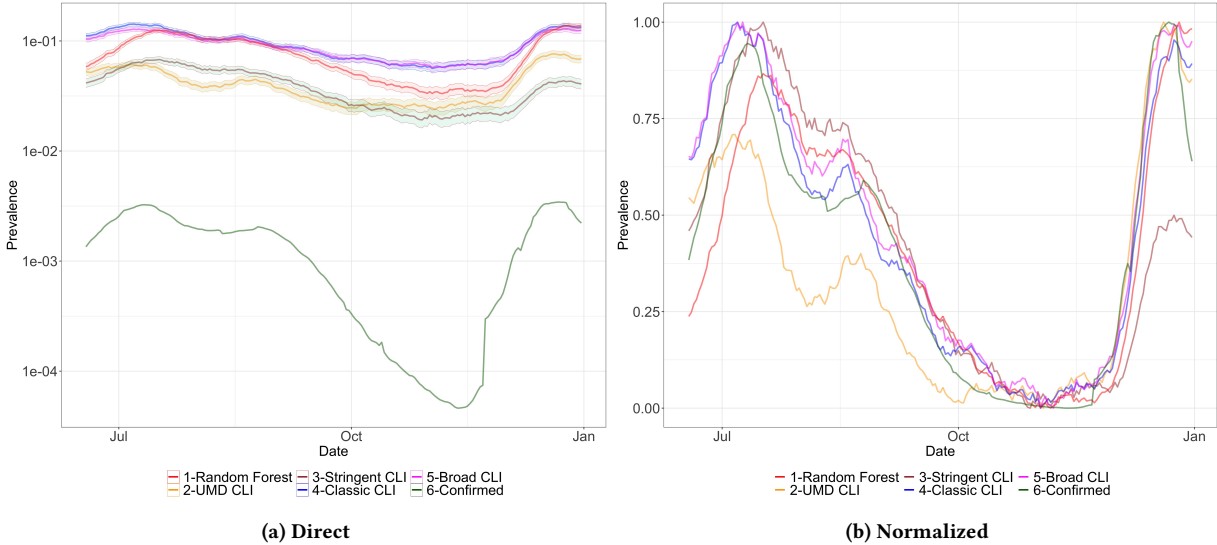

**(a) Direct**                    **(b) Normalized**

**Figure 1: Prevalence in South Africa obtained with the different proxies, smoothed with a rolling average of 14 days from June 18th to December 31st, 2021. In plot (a) we have the actual ratio (note that the y axis is in logarithmic scale). In plot (b) all curves are normalized so the smallest value is 0 and the largest value is 1.**

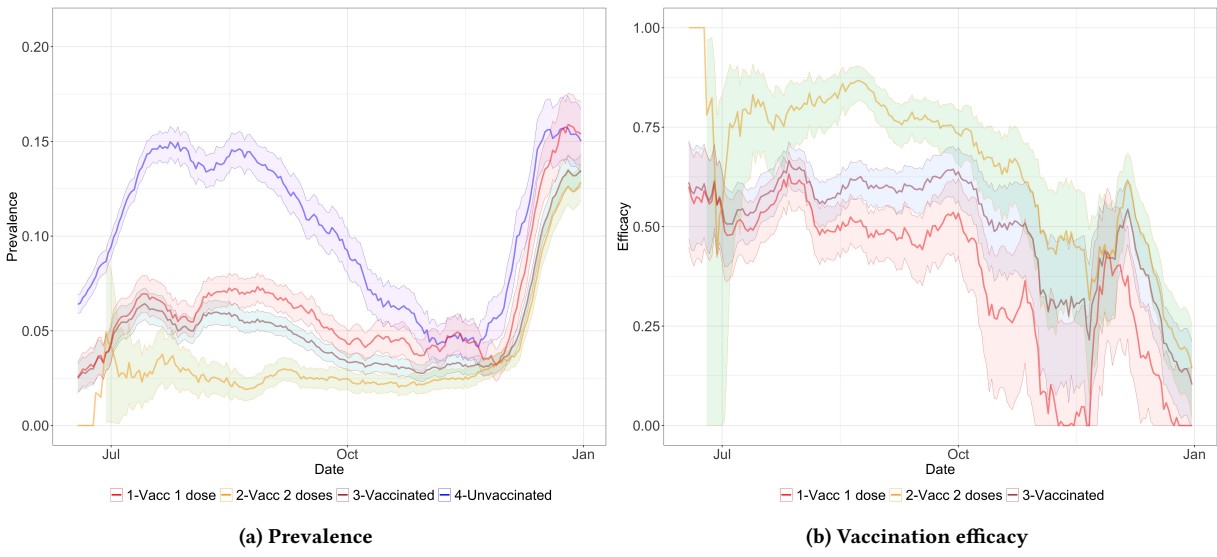

**(a) Prevalence**                    **(b) Vaccination efficacy**

**Figure 2: Prevalence (a) and vaccination efficacy (b) in South Africa among people with different levels of vaccination, estimated with Random Forest.**

## 2.4 Countries and Time Periods

*2.4.1 South Africa.* The main objective of this work is to evaluate the change in vaccine efficacy due to the Omicron variant. To this end, we evaluate the decrease in vaccine efficacy in South Africa and the Gauteng province, which is among the most affected, from mid-June 2021 until the end of 2021. Moreover, to ensure that we have sufficient data for our estimates, we concentrate on three recent time periods, each lasting about a month, where more data is available. During two of these time periods the Delta variant is

dominant: i) June 18 to July 18, 2021, the period considered in [31] with low vaccination level, and ii) August 9 to September 6, 2021; while in the last time period, December 1st to 31st, 2021, Omicron is dominant (the information on variant presence is obtained from [22], which extracts it from [7] via [10], details are available Table 5).

*2.4.2 World.* Beyond South Africa, we study the 50 countries for which the UMD Global CTIS has the largest amount of data. For all of them we compute the vaccine efficacy in the month of October (in which Omicron was still not present) and in the month of December

(in which Omicron was present). A computed efficacy value is only considered if i) it is non-negative, ii) both prevalences $P_V$ and $P_U$ are at least 0.01, and iii) the number of samples used to compute them is at least 1000. We only consider further those countries for which these three conditions hold for the efficacy value in December of at least one among the vaccination status cases we consider.

We have observed that the information on prevalence of Omicron becomes available [22] with a significant delay. Hence, most countries do not report relevant presence of Omicron until the second half of December 2021. For that reason, we consider the prevalence of Omicron reported from December 15th, 2021 to January 7th, 2022. Furthermore, among the countries mentioned above, in order to have a reasonable estimate of the prevalence of the Omicron variant, we consider only countries whose data is based on sequencing at least 30 virus samples. We say that these are the countries with *presence of Omicron*. For all countries with presence of Omicron, we compare the estimated vaccination efficacy using Random Forest among all three vaccination groups and for both periods. For this, we adopt simple statistical methods, such as correlation analysis.

## 3 RESULTS

### 3.1 Prevalence and Vaccination Efficacy in South Africa

Figures 1a and 1b show the prevalence of COVID-19 in South Africa in the period June 18th to December 31st, 2021, with the different proxies. The direct approach of Figure 1a shows a gap between the estimate Confirmed derived from the official number of cases and the other proxies. This gap can be explained in part by under-detection in the official number of cases (in South Africa the test-positivity rate is above 15%, as seen in Table 2). More generally (in South Africa and elsewhere) symptom-based proxies can overestimate the number of cases when respondents report symptoms that are consistent with COVID-19 but are produced by some other condition. Figure 1b shows that if each curve is independently normalized to the unit scale, all proxies closely track the evolution of the official number of cases Confirmed.

We also analyzed the prevalence of COVID-19 in South Africa depending on the vaccination status with the different proxies. Our results (see Figures 5a to 5d in the Appendix) indicate that the UMD CLI and Stringent CLI proxies show a low infection prevalence in July-September and December when compared with the other proxies. On the other hand, Classic CLI and Broad CLI show a high prevalence in the period October-November, when the official data was showing that the number of cases was very low, possibly because of existing symptoms in the population not related to COVID-19.

Here we focus on the Random Forest proxy; Figure 2a shows the prevalence in South Africa across all reported vaccination states. We can observe that the magnitude of the two waves (August-September and December) is similar among the Unvaccinated population, while in the vaccinated groups (Vaccinated, Vaccinated with 1 dose and Vaccinated with 2 doses) there is a much higher rate of prevalence in the December wave. This hints at a decrease of vaccine efficacy towards infection with the introduction of Omicron, as we will show next. We also observe that, as expected, subjects

vaccinated with two doses show higher protection that those reporting only one dose (with Vaccinated somewhere in between since it combines both groups).

As for vaccination efficacy, Figure 2b shows the estimates for South Africa, again with Random Forest. The estimates in October-November have lower quality due to the sharp reduction of cases in that period. However, when contrasting the August-September period (with mostly Delta presence) to the December period (with mostly Omicron presence) we can clearly observe the reduction of vaccine efficacy towards infection in the latter.

Table 3 quantifies the estimated efficacy for the three periods of interest and for the five classifiers, for South Africa and for the Gauteng province.

### 3.2 Prevalence and Vaccination Efficacy in the World

From the 50 countries with the largest amount of data in the CTIS and having *presence of Omicron*, we select those with an acceptable estimated efficacy value (where estimates are accepted if they follow the three rules listed in Section 2.4.2). This results in a set of 24 countries. Table 4 presents the estimates of virus prevalence in the these countries in the periods of October and December, and also estimates of vaccination efficacy towards infection. Details about vaccination levels are presented in Table 8 (vaccination data is obtained from [19, 22]; the different vaccine types used in different countries have not been taken into account).

Both prevalence estimates and the derived efficacy estimates are obtained by the Random Forest classifier and shown with 95% confidence intervals. The left-hand side of Table 4 focuses on the data from individuals that declared their overall vaccination status (using groups Vaccinated, Unvaccinated); its right-hand side makes a more detailed characterization by considering the number of doses declared (groups Vaccinated with 1 dose, Vaccinated with 2 doses, Unvaccinated). We also observe that there is less data on individuals with only one dose, since this is a transient state in the vaccination sequence. The full information on sample sizes can be consulted in Tables 9 and 10.

Figure 3 complements the data in the tables. Figure 3a shows three pairs of box plots. Each pair allows comparing vaccine efficacy in October and December when considering data from the selected countries. We observe that although results are inconclusive for Vaccinated with 1 dose, there is a clear decrease of overall efficacy when considering Vaccinated and Vaccinated with 2 doses. Tabular data is available in Table 6.

Figures 3b-3d allow us to see a clear trend when plotting efficacy against the most recent relative level of Omicron presence in each selected country. For each case, we present a smoothed line (Loess fitting curve, in blue), depicting a clear decreasing trend. We also evaluated the correlation coefficient (using Pearson correlation) and the corresponding p-value, which confirms its statistical significance for the usual $\alpha = 5\%$ (details in Table 7).

## 4 DISCUSSION

After its surge in South Africa, the Omicron variant is increasing in prevalence in other countries. Although it is still unclear if this variant is associated to a milder disease [13] several studies have

| Method | Jun-Jul Efficacy [95%CI] | Aug-Sep Efficacy [95%CI] | Dec Efficacy [95%CI] | Jun-Jul Efficacy [95%CI] | Aug-Sep Efficacy [95%CI] | Dec Efficacy [95%CI] |
|---|---|---|---|---|---|---|
| | South Africa | | | Gauteng | | |
| | Vaccinated | | | | | |
| Random Forest | 0.54 [0.48,0.59] | 0.62 [0.58,0.65] | 0.24 [0.17,0.30] | 0.43 [0.33,0.51] | 0.62 [0.54,0.69] | 0.30 [0.18,0.40] |
| UMD CLI | 0.60 [0.53,0.66] | 0.66 [0.61,0.70] | 0.46 [0.39,0.51] | 0.58 [0.44,0.68] | 0.63 [0.51,0.73] | 0.52 [0.41,0.61] |
| Stringent CLI | 0.69 [0.63,0.74] | 0.70 [0.66,0.73] | 0.48 [0.40,0.55] | 0.64 [0.53,0.72] | 0.70 [0.61,0.78] | 0.57 [0.43,0.67] |
| Classic CLI | 0.55 [0.50,0.59] | 0.56 [0.52,0.59] | 0.38 [0.33,0.43] | 0.50 [0.42,0.58] | 0.51 [0.42,0.59] | 0.48 [0.39,0.55] |
| Broad CLI | 0.50 [0.44,0.54] | 0.49 [0.44,0.52] | 0.36 [0.30,0.41] | 0.49 [0.39,0.57] | 0.41 [0.31,0.50] | 0.45 [0.35,0.53] |
| | Vaccinated with one dose | | | | | |
| Random Forest | 0.50 [0.44,0.56] | 0.51 [0.46,0.55] | 0.09 [0.00,0.18] | 0.40 [0.28,0.49] | 0.54 [0.44,0.63] | 0.14 [0.00,0.30] |
| UMD CLI | 0.61 [0.54,0.68] | 0.56 [0.50,0.62] | 0.21 [0.09,0.31] | 0.60 [0.46,0.71] | 0.58 [0.42,0.70] | 0.38 [0.18,0.53] |
| Stringent CLI | 0.67 [0.61,0.73] | 0.60 [0.54,0.65] | 0.23 [0.07,0.36] | 0.62 [0.49,0.71] | 0.61 [0.47,0.71] | 0.39 [0.13,0.57] |
| Classic CLI | 0.53 [0.47,0.57] | 0.47 [0.42,0.51] | 0.21 [0.13,0.28] | 0.47 [0.37,0.56] | 0.47 [0.36,0.56] | 0.35 [0.20,0.46] |
| Broad CLI | 0.46 [0.40,0.52] | 0.39 [0.34,0.44] | 0.18 [0.09,0.26] | 0.44 [0.33,0.53] | 0.34 [0.20,0.45] | 0.29 [0.14,0.42] |
| | Vaccinated with two doses | | | | | |
| Random Forest | 0.76 [0.64,0.84] | 0.81 [0.78,0.84] | 0.30 [0.23,0.36] | 0.62 [0.36,0.78] | 0.77 [0.67,0.85] | 0.36 [0.24,0.46] |
| UMD CLI | 0.75 [0.57,0.86] | 0.85 [0.79,0.88] | 0.56 [0.50,0.61] | 0.69 [0.27,0.87] | 0.73 [0.54,0.84] | 0.57 [0.45,0.66] |
| Stringent CLI | 0.82 [0.66,0.90] | 0.88 [0.84,0.91] | 0.59 [0.51,0.65] | 0.85 [0.55,0.95] | 0.88 [0.76,0.94] | 0.65 [0.51,0.74] |
| Classic CLI | 0.77 [0.66,0.84] | 0.71 [0.67,0.75] | 0.45 [0.40,0.49] | 0.79 [0.59,0.90] | 0.58 [0.44,0.68] | 0.53 [0.44,0.60] |
| Broad CLI | 0.75 [0.63,0.83] | 0.66 [0.61,0.71] | 0.43 [0.37,0.48] | 0.80 [0.59,0.91] | 0.54 [0.39,0.65] | 0.50 [0.41,0.58] |

**Table 3: Vaccine efficacy in South Africa and the Gauteng province, calculated for three time periods: June 18th to July 18th (Jun-Jul), August 9th to September 6th (Aug-Sep), and December 1st to 31st (Dec).**

| Country | Prevalence Omicron | Prevalence COVID-19 Oct | Prevalence COVID-19 Dec | Vac efficacy Oct | Vac efficacy Dec | Vac 1 dose efficacy Oct | Vac 1 dose efficacy Dec | Vac 2 doses efficacy Oct | Vac 2 doses efficacy Dec |
|---|---|---|---|---|---|---|---|---|---|
| Argentina | 0.83 [0.76,0.91] | 0.02 [0.01,0.02] | 0.03 [0.03,0.03] | 0.48 [0.35,0.58] | 0.28 [0.12,0.41] | 0.03 [0.00,0.27] | – | 0.53 [0.41,0.62] | 0.31 [0.15,0.43] |
| Belgium | 0.32 [0.29,0.34] | 0.02 [0.02,0.02] | 0.05 [0.05,0.05] | 0.53 [0.39,0.64] | 0.38 [0.26,0.48] | – | – | 0.55 [0.41,0.65] | 0.38 [0.26,0.48] |
| Brazil | 0.58 [0.52,0.64] | 0.03 [0.03,0.03] | 0.03 [0.02,0.03] | 0.43 [0.37,0.49] | 0.29 [0.19,0.38] | 0.20 [0.11,0.28] | – | 0.50 [0.44,0.55] | 0.33 [0.23,0.41] |
| Colombia | 0.35 [0.26,0.44] | 0.03 [0.03,0.03] | 0.03 [0.03,0.03] | 0.55 [0.49,0.61] | 0.49 [0.39,0.56] | 0.44 [0.35,0.53] | 0.36 [0.22,0.47] | 0.61 [0.55,0.67] | 0.53 [0.45,0.61] |
| Denmark | 0.47 [0.46,0.49] | 0.01 [0.01,0.01] | 0.05 [0.05,0.05] | – | 0.49 [0.39,0.57] | – | – | – | 0.48 [0.38,0.57] |
| France | 0.26 [0.24,0.27] | 0.01 [0.01,0.01] | 0.03 [0.03,0.03] | – | 0.44 [0.39,0.49] | – | 0.46 [0.35,0.55] | – | 0.44 [0.39,0.49] |
| Germany | 0.13 [0.13,0.14] | 0.01 [0.01,0.01] | 0.02 [0.02,0.02] | – | 0.65 [0.62,0.68] | – | 0.44 [0.34,0.53] | – | 0.66 [0.63,0.69] |
| India | 0.33 [0.29,0.38] | 0.04 [0.04,0.04] | 0.03 [0.03,0.03] | 0.44 [0.35,0.52] | 0.42 [0.28,0.53] | 0.19 [0.05,0.31] | 0.07 [0.00,0.26] | 0.54 [0.47,0.61] | 0.49 [0.37,0.58] |
| Italy | 0.21 [0.19,0.22] | 0.01 [0.01,0.01] | 0.02 [0.02,0.02] | – | 0.61 [0.57,0.65] | – | 0.66 [0.57,0.72] | – | 0.61 [0.56,0.65] |
| Mexico | 0.54 [0.49,0.58] | 0.05 [0.05,0.05] | 0.04 [0.04,0.04] | 0.57 [0.54,0.59] | 0.51 [0.46,0.55] | 0.36 [0.32,0.40] | 0.22 [0.14,0.30] | 0.66 [0.63,0.68] | 0.56 [0.52,0.60] |
| Netherlands | 0.30 [0.27,0.33] | 0.02 [0.02,0.02] | 0.05 [0.04,0.05] | 0.36 [0.20,0.49] | 0.29 [0.18,0.38] | – | 0.16 [0.00,0.33] | 0.41 [0.26,0.53] | 0.30 [0.19,0.39] |
| Norway | 0.25 [0.15,0.36] | 0.01 [0.01,0.01] | 0.03 [0.02,0.03] | – | 0.35 [0.10,0.52] | – | – | – | 0.35 [0.11,0.53] |
| Poland | 0.03 [0.02,0.04] | 0.03 [0.03,0.04] | 0.07 [0.06,0.07] | 0.50 [0.42,0.56] | 0.57 [0.53,0.60] | 0.31 [0.13,0.45] | 0.44 [0.34,0.52] | 0.52 [0.45,0.58] | 0.58 [0.55,0.62] |
| Portugal | 0.23 [0.19,0.27] | 0.01 [0.01,0.01] | 0.03 [0.03,0.03] | – | 0.32 [0.12,0.48] | – | 0.23 [0.00,0.44] | – | 0.33 [0.13,0.49] |
| Romania | 0.04 [0.00,0.08] | 0.06 [0.06,0.06] | 0.02 [0.02,0.02] | 0.59 [0.56,0.62] | 0.65 [0.57,0.71] | 0.65 [0.59,0.70] | 0.52 [0.33,0.65] | 0.58 [0.55,0.61] | 0.68 [0.60,0.74] |
| Russia | 0.29 [0.22,0.36] | 0.04 [0.04,0.05] | 0.03 [0.02,0.03] | 0.45 [0.39,0.50] | 0.43 [0.34,0.51] | 0.55 [0.43,0.64] | 0.30 [0.09,0.46] | 0.44 [0.38,0.50] | 0.46 [0.37,0.53] |
| Slovakia | 0.10 [0.03,0.17] | 0.03 [0.03,0.03] | 0.06 [0.05,0.06] | 0.47 [0.32,0.59] | 0.54 [0.46,0.61] | – | – | 0.50 [0.35,0.61] | 0.55 [0.47,0.62] |
| South Africa | 0.88 [0.81,0.96] | 0.04 [0.04,0.04] | 0.12 [0.12,0.13] | 0.50 [0.41,0.57] | 0.24 [0.17,0.30] | 0.29 [0.15,0.40] | 0.09 [0.00,0.18] | 0.64 [0.56,0.70] | 0.30 [0.23,0.36] |
| Spain | 0.46 [0.43,0.50] | 0.01 [0.01,0.02] | 0.05 [0.05,0.06] | 0.62 [0.50,0.70] | 0.26 [0.15,0.36] | 0.34 [0.09,0.52] | 0.30 [0.15,0.43] | 0.66 [0.55,0.74] | 0.26 [0.14,0.36] |
| Sweden | 0.34 [0.32,0.37] | 0.01 [0.00,0.01] | 0.02 [0.02,0.02] | – | 0.48 [0.36,0.57] | – | – | – | 0.48 [0.36,0.57] |
| Switzerland | 0.39 [0.36,0.41] | 0.01 [0.01,0.01] | 0.04 [0.04,0.04] | – | 0.52 [0.43,0.59] | – | – | – | 0.51 [0.42,0.59] |
| Turkey | 0.10 [0.08,0.11] | 0.05 [0.05,0.06] | 0.05 [0.05,0.05] | 0.45 [0.38,0.51] | 0.42 [0.33,0.51] | – | – | 0.49 [0.42,0.55] | 0.44 [0.34,0.52] |
| United Kingdom | 0.66 [0.65,0.66] | 0.03 [0.03,0.03] | 0.05 [0.04,0.05] | 0.34 [0.22,0.45] | 0.20 [0.07,0.31] | – | – | 0.36 [0.24,0.46] | 0.21 [0.08,0.32] |
| Vietnam | 0.02 [0.00,0.06] | 0.01 [0.01,0.01] | 0.03 [0.03,0.03] | – | – | – | 0.25 [0.00,0.50] | – | – |

**Table 4: Prevalence of Omicron, prevalence of COVID-19, and vaccination efficacy in the countries with presence of Omicron (as defined in Section 2.4.2). When data is insufficient to meet the defined selection criteria, it is omitted and replaced by "–".**

raised concerns over the decrease of vaccine effectiveness against infection [14, 18, 20, 23] and this can lead to a wider spread of the virus even in countries with a high vaccination uptake.

Daily participatory symptom surveillance has the potential to offer a new instrument for assessing both global and local trends in health status. While limited in assessing the ground truth, due to the smaller control over the sample design and the need to preserve anonymity, we believe that the vast number of daily survey responses can compensate some of these factors. In this study, we developed a method to adapt and calibrate against the reported SARS-CoV-2 infection status the selection of symptoms, and other covariates from the survey, along different time periods and locations. As compared to methods that only use the presence or absence of symptoms reported by survey respondents [31], our proposed method was shown to provide a better proxy for assessing the trend in infections, more closely tracking the official reported cases, in particular in those countries that had a strong surveillance and consistent test positivity rates.

Using this improved classifier we complemented earlier results [31] that used traditional fixed combinations of symptoms, and

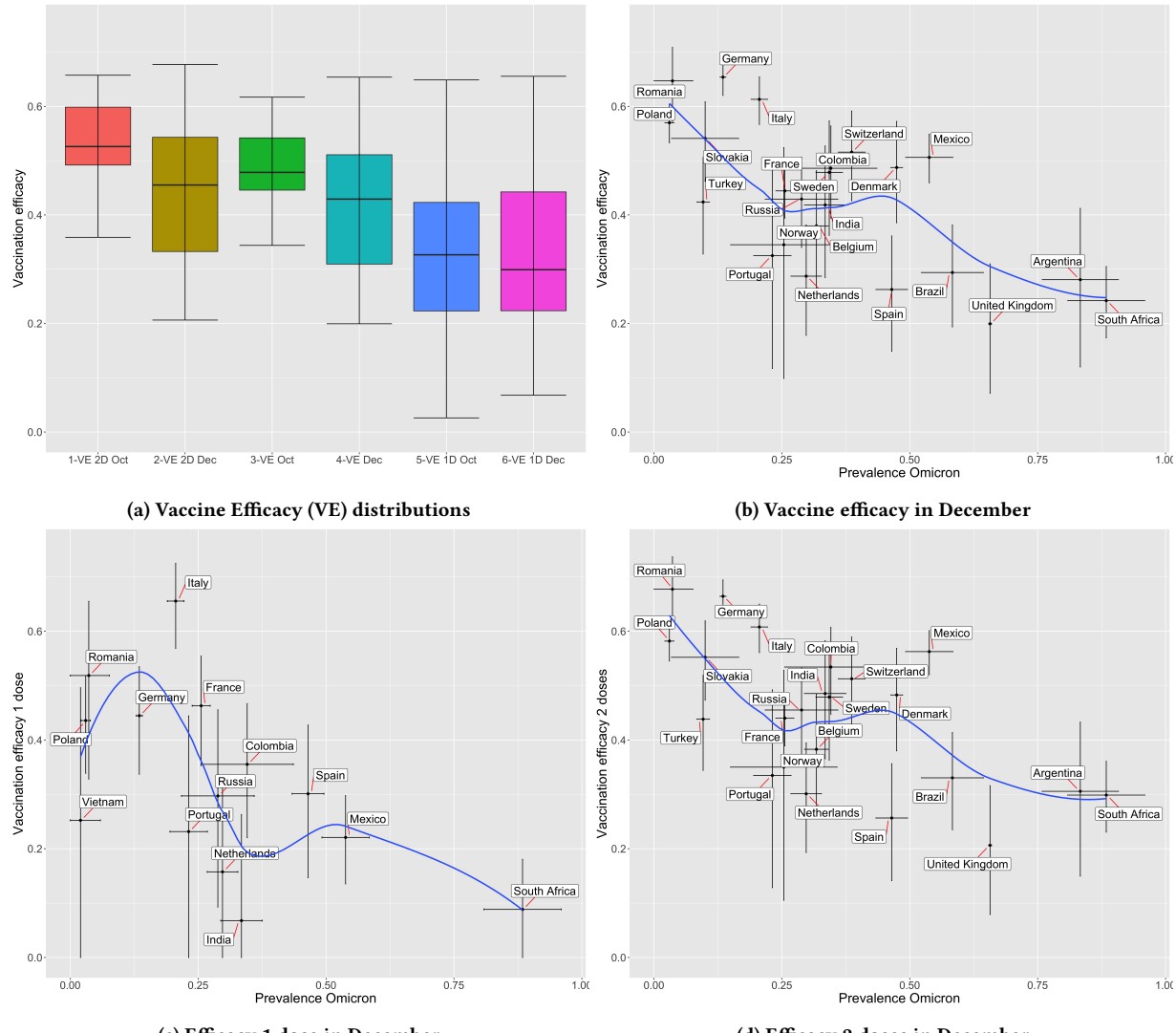

(a) Vaccine Efficacy (VE) distributions

(b) Vaccine efficacy in December

(c) Efficacy 1 dose in December

(d) Efficacy 2 doses in December

Figure 3: Analysis of vaccine efficacy towards preventing infection: Sub-figure (a) shows distributions of efficacy in October and December, for the countries with presence of Omicron (as defined in Section 2.4.2); Sub-figures (b,c,d) show vaccination efficacy versus Omicron prevalence in the same set of countries, depending on vaccination status. For each country the 95% confidence intervals of the two values are shown as black lines. The blue line is the Loess curve fitting of the data.

updated the analysis for South Africa showing the observed decrease in vaccine efficacy when contrasting a Delta-dominated period (August-September 2021) with the recent Omicron-dominated period (December 2021). We confirmed the presence of a measurable drop in vaccine efficacy from 0.62 (with 95% confidence interval [0.58, 0.65]) in the Delta period to 0.24 (95% CI [0.17, 0.30]) in the Omicron period in the whole country (0.62[0.54, 0.69] to 0.30[0.18, 0.40] in the Gauteng province). In addition, we confirmed that having two doses of vaccine confers better protection than one dose, both in Delta (0.81[0.78, 0.84] versus 0.51[0.46, 0.55]) and Omicron (0.30[0.23, 0.36] versus 0.09[0.00, 0.18]) dominated periods. However, we have no data on the status of respondents with regard to a possible booster dose.

These results are in line with other studies on the vaccine effectiveness against infection and more severe outcomes conducted in several countries. Andrews et al. [4] used data from England to study vaccine effectiveness against symptomatic infection. They found that vaccine effectiveness against symptomatic disease was higher for the Delta variant than for the Omicron variant. They found that two doses of the ChAdOx1 nCoV-19 or the BNT162b2 vaccine provided low protection against Omicron, and that, while a BNT162b2 or mRNA-1273 booster increased this protection, it also waned with time. Buchan et al. [6] use data from Ontario, Canada, to find out that vaccine effectiveness is higher with Delta than with Omicron. For instance, 7-59 days after a second dose, effectiveness against symptomatic infection was estimated to be

89% for Delta and only 36% for Omicron. After a third dose, effectiveness against symptomatic infection increased to 97% an 61%, respectively, and was more than 95% against severe outcomes for both variants. Kodera et al. [15] estimate the effectiveness of vaccination against Delta and Omicron in Japan, and how it decreases with time (the waning immunity). They found that the effectiveness of vaccination for the Delta variant was 95% after the second shot. From the reported data of 25,187 positive cases with confirmed Omicron variant in Tokyo in January 2022, the effectiveness of vaccination against Omicron after the second dose is estimated below 65% compared to that of the Delta variant. Hansen et al. [9] use Danish data to report vaccination effectiveness against Omicron (B.1.1.529). They report effectiveness of 55.2% and 36.7% with the BNT162b2 and mRNA-1273 vaccines, respectively, in the first month after primary vaccination. They also observe that this effectiveness is significantly lower than that against Delta infection and declines rapidly. The effectiveness is re-established (54.6%) with a booster shot of the BNT162b2 vaccine. So, all these studies find that a difference in the effectiveness against infection of Omicron versus Delta.

A difference can also be observed in the effectiveness against hospitalization. Lauring et al. [16] evaluate effectiveness of mRNA vaccines. They found that they are highly effective preventing COVID-19 associated hospital admissions against Delta: 85% with two doses and 94% with three doses, while the effectiveness decreases against Omicron: 65% with two doses and 86% with three doses. The effectiveness against in-hospital mortality is also higher against Delta (12.2%) than Omicron (7.1%). Stowe et al. [28] use data from England to compare vaccine effectiveness against hospitalization with Delta and Omicron. They observe that, while effectiveness is lower and waning is faster for Omicron, this is partially due to incidental cases (hospitalizations not caused by COVID-19), and the difference decreases when only severe respiratory hospitalizations are considered.

By January 7th, 2022, when we completed this study, there was a small number of candidate countries exhibiting both a high prevalence of Omicron and a high level of sequencing data supporting it. Nevertheless, we extend our analysis to these countries and show the observed changes in efficacy when comparing the months of October (pre-Omicron) with December (with partial presence of Omicron). Although these results should be confirmed once the level of Omicron becomes more dominant in many countries, we have observed a significant level of correlation of around and beyond −0.6 between vaccine efficacy (with either one or two doses) and the prevalence of Omicron. We must also make it clear that our results show a reduction in efficacy in terms of protection against infection, but this does not imply a reduction of vaccine efficacy in protection against serious disease, hospitalization and death.

There are several assumptions that frame our analysis. We assume that UMD Global CTIS answers provide a sample of the population that is interchangeable among the Delta and Omicron dominated periods. Additionally, we did not take into account possible effects from waning immunity and vaccine boost shots. However, within the countries we consider we have a mix of different vaccination timings, so that our observations appear to be valid under different scenarios. We leave for future work a further analysis where vaccination timing is taken into account.

## ACKNOWLEDGMENTS

Work supported by the Comunidad de Madrid under Grants COMODIN-CM, supported with REACT-EU funds, and CoronaSurveys-CM.

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

## A  ETHICAL DECLARATION

The Ethics Board (IRB) of IMDEA Networks Institute gave ethical approval for this work on 2021/07/05. IMDEA Networks has signed Data Use Agreements with Facebook, Carnegie Mellon University (CMU) and the University of Maryland (UMD) to access their data, specifically UMD project 1587016-3 entitled C-SPEC: Symptom Survey: COVID-19 and CMU project STUDY2020_00000162 entitled ILI Community-Surveillance Study. The data used in this study was collected by the University of Maryland through The University of Maryland Social Data Science Center Global COVID-19 Trends and Impact Survey in partnership with Facebook. Informed consent has been obtained from all participants in this survey by this institution (see [29]). All the methods in this study have been carried out in accordance with relevant of ethics and privacy guidelines and regulations.

## B  AVAILABILITY OF DATA AND MATERIALS

The data presented in this paper (in aggregated form) and the programs used to process it are openly accessible at https://github.com/GCGImdea/coronasurveys/tree/master/papers/omicron_efficacy_paper_medRxiv. The microdata of the CTIS survey from which the aggregated data was obtained cannot be shared, as per the Data Use Agreements signed with Facebook, Carnegie Mellon University (CMU) and the University of Maryland (UMD).

## C  LIST OF SYMPTOMS

In the UMD Global CTIS the following question is asked: "B1: In the last 24 hours, have you had any of the following?" [29]. The following is the list of possible answers (non exclusive): Fever (B1_1), Cough (B1_2), Difficulty breathing (B1_3), Fatigue (B1_4), Stuffy or runny nose (B1_5), Aches or muscle pain (B1_6), Sore throat (B1_7), Chest pain (B1_8), Nausea (B1_9), Loss of smell or taste (B1_10), Headache (B1_12), Chills (B1_13).

## D  QUESTIONS USED FOR THE MACHINE LEARNING MODEL

The following is the list of survey questions whose answers are used to create the Random Forest models, and to classify with them the responses: B1_1, B1_2, B1_3, B1_4, B1_5, B1_6, B1_7, B1_8, B1_9, B1_10, B1_11, B1_12, B1_13, B1_14, B1b_x1, B1b_x2, B1b_x3, B1b_x4, B1b_x5, B1b_x6, B1b_x7, B1b_x8, B1b_x9, B1b_x10, B1b_x11, B1b_x12, B1b_x13, B1b_x14, B3, B5, B6, B9, B10, B11, B12_1, B12_2, B12_3, B12_4, B12_5, B12_6, B13_1, B13_2, B13_3, B13_4, B13_5, B13_6, B13_7, B14_1, B14_2, B14_3, B14_4, B14_5, C0_1, C0_2, C0_3, C0_4, C0_5, C0_6, C1_m, C2, C3, C5, C6, C7, C8, C9, C9a, C12, C13_1, C13_2, C13_3, C13_4, C13_5, C13_6, C14, D1, D2, D3, D4, D5, D6_1, D6_2, D6_3, D7, D8, D9, D10, E2, E3, E4, E7, H1, H2, H3.

The questions removed are B0, B7, B8, B15, and all the questions related to vaccination (V-questions).

## E  VACCINATION IN SOUTH AFRICA

Figure 4 shows an area plot, estimated from the UMD Global CTIS data, of the proportion of vaccinated with 1 dose, Vaccinated with 2 doses, and Unvaccinated from June 18th until December 31st, 2021. As can be seen, the ratio of the population vaccinated is low at the beginning of this interval, especially with two doses. Then, we can see a high increase in Vaccinated between July and October. We point out that in each time point of this plot the proportions are provided by a different set of surveys respondents, and it still closely captures the increase of vaccination.

## F  COUNTRIES WITH OMICRON PREVALENCE

Table 8 shows basic official vaccination data on December 31st, 2021, of these countries. Table 4 shows the COVID-19 prevalence and the vaccine efficacy in October and December in the countries with presence of Omicron as defined in Section 2.4.2. When data is insufficient to meet the defined selection criteria, it is omitted and replaced by "−". Both tables are presented alphabetically by country name and also share a column depicting the most recent data on Omicron prevalence among all virus samples.

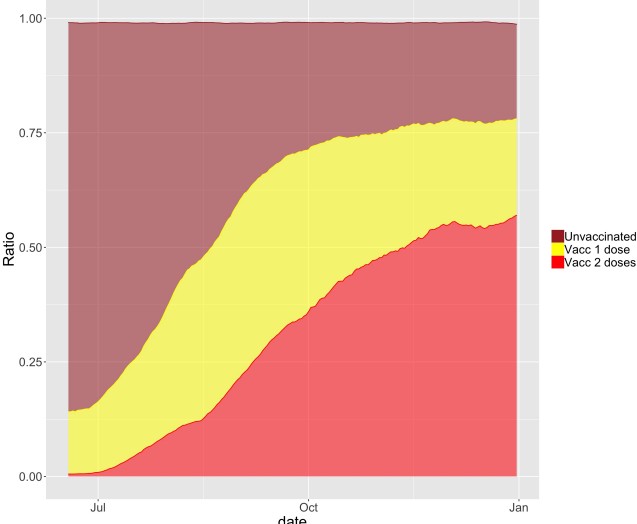

**Figure 4: Evolution of the vaccination in South Africa as ratio of the population, estimated from the UMD Global CTIS data. A small fraction of responses that declared being vaccinated without reporting the number of doses are not presented for clarity. The values are from June 18th to December 31st, 2021, smoothed with a rolling average of 14 days.**

| Date | % Delta | % Omicron | # samples |
|---|---|---|---|
| 2021-06-14 | 45.23 | 0.00 | 1101 |
| 2021-06-28 | 78.09 | 0.00 | 1661 |
| 2021-07-12 | 88.90 | 0.00 | 2226 |
| 2021-07-26 | 94.30 | 0.00 | 1667 |
| 2021-08-09 | 95.19 | 0.00 | 1601 |
| 2021-08-23 | 97.58 | 0.00 | 1242 |
| 2021-09-06 | 97.01 | 0.00 | 1269 |
| 2021-09-20 | 95.77 | 0.00 | 923 |
| 2021-10-04 | 93.57 | 0.00 | 513 |
| 2021-10-18 | 93.56 | 0.00 | 450 |
| 2021-11-01 | 95.67 | 0.48 | 208 |
| 2021-11-15 | 69.30 | 20.18 | 114 |
| 2021-11-29 | 13.08 | 85.00 | 780 |
| 2021-12-13 | 0.92 | 95.92 | 980 |
| 2021-12-27 | 0.00 | 93.85 | 65 |

**Table 5: Percentage of sequenced virus samples belonging to Delta and Omicron in South Africa from June 1st to December 31st of 2021. The third column presents the total number of samples reported on the corresponding date.**

| Vaccination status | Prevalence | | Vaccination efficacy | |
|---|---|---|---|---|
| | October | December | October | December |
| Vaccinated 2 doses | 0.02 [0.01,0.02] | 0.03 [0.03,0.04] | 0.53 [0.49,0.58] | 0.45 [0.39,0.50] |
| Vaccinated | 0.02 [0.01,0.03] | 0.04 [0.03,0.04] | 0.49 [0.45,0.52] | 0.43 [0.37,0.48] |
| Vaccinated 1 dose | 0.03 [0.02,0.04] | 0.05 [0.04,0.06] | 0.34 [0.22,0.45] | 0.32 [0.23,0.41] |
| Unvaccinated | 0.04 [0.03,0.05] | 0.06 [0.05,0.07] | – | – |

**Table 6: Prevalence of COVID-19 and vaccine efficacy (with 95% confidence interval) in the countries with presence of Omicron in the periods of October and December 2021.**

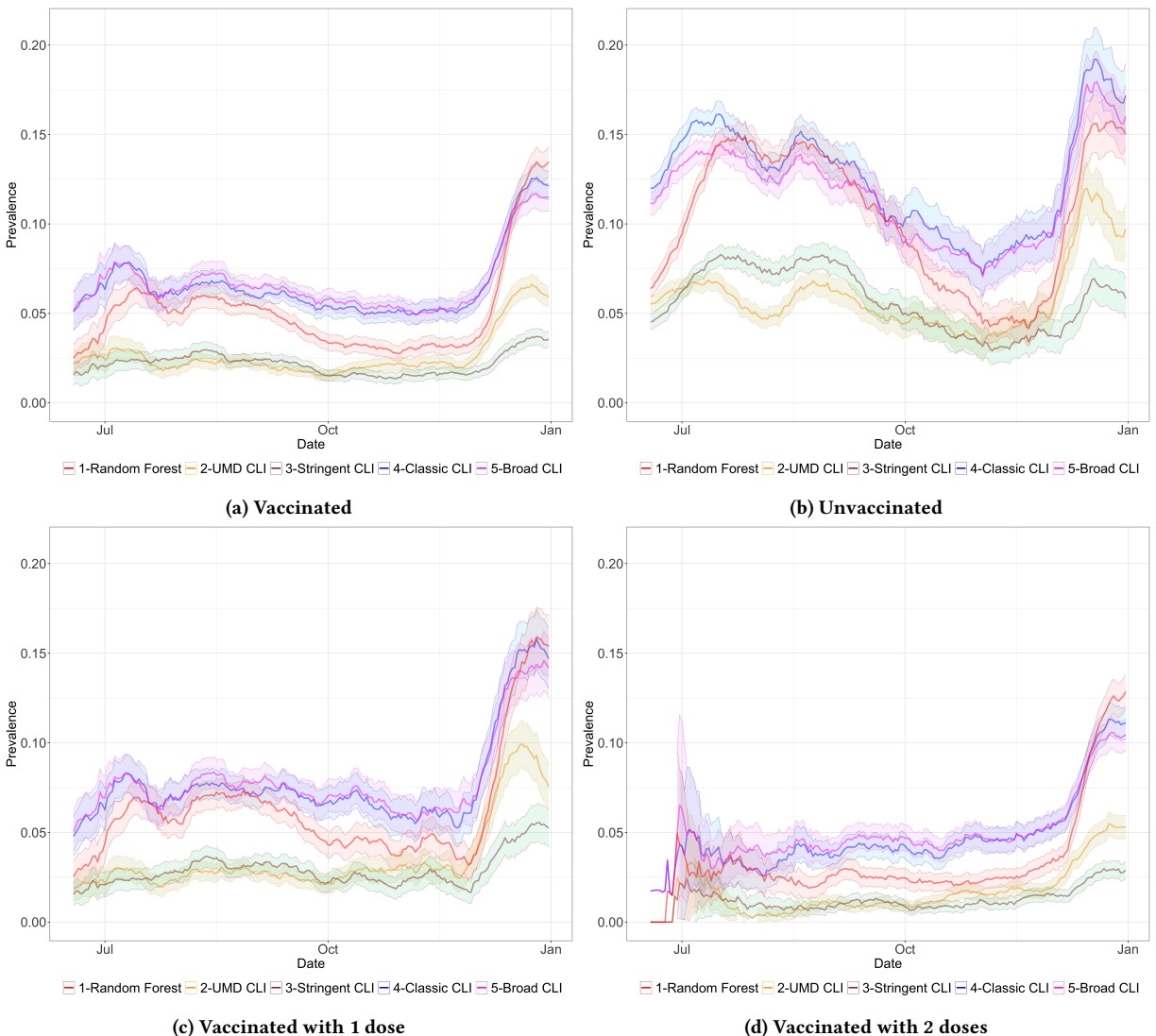

(a) Vaccinated

(b) Unvaccinated

(c) Vaccinated with 1 dose

(d) Vaccinated with 2 doses

**Figure 5: Prevalence in South Africa among Vaccinated, Unvaccinated, Vaccinated with 1 dose, and Vaccinated with 2 doses, with different proxies.**

| Prevalence omicron vs | Correlation coefficient | P-value |
|---|---|---|
| Vaccination efficacy | -0.680301 | 0.000354 |
| Vacc. efficacy 1 dose | -0.564977 | 0.035274 |
| Vacc. efficacy 2 doses | -0.628936 | 0.001306 |

**Table 7: Correlation between prevalence of Omicron and vaccine efficacy in the countries with presence of Omicron.**

| Country | % doses/pop | % pop vacc | % pop fully vacc | % pop booster | Vacc start date |
|---|---|---|---|---|---|
| Argentina | 167.98 | 83.76 | 71.61 | 12.22 | 2020-12-29 |
| Belgium | 186.28 | 76.65 | 75.70 | 37.59 | 2020-12-28 |
| Brazil | 154.81 | 77.66 | 67.03 | 12.42 | 2021-01-17 |
| Colombia | 126.19 | 74.81 | 55.25 | 6.49 | 2021-02-17 |
| Denmark | 208.57 | 82.65 | 78.43 | 48.30 | 2021-02-05 |
| France | 183.78 | 78.61 | 73.48 | 33.28 | 2020-12-27 |
| Germany | 178.84 | 73.62 | 70.61 | 38.87 | 2020-12-27 |
| India | 103.98 | 60.69 | 43.29 | 0.00 | 2021-01-16 |
| Italy | 184.28 | 80.14 | 74.11 | 32.52 | 2020-12-27 |
| Mexico | 114.24 | 62.89 | 55.87 | 0.00 | 2020-12-24 |
| Netherlands | 162.18 | 77.54 | 71.18 | 18.50 | 2021-01-09 |
| Norway | 178.68 | 78.41 | 71.76 | 28.52 | 2020-12-08 |
| Poland | 124.32 | 57.34 | 55.68 | 18.16 | 2020-12-28 |
| Portugal | 190.72 | 91.47 | 89.53 | 29.44 | 2020-12-27 |
| Romania | 82.86 | 28.64 | 40.87 | 0.00 | 2020-12-27 |
| Russia | 100.31 | 50.60 | 45.76 | 5.06 | 2020-12-15 |
| Slovakia | 111.09 | 50.13 | 47.61 | 16.33 | 2021-01-11 |
| South Africa | 46.47 | 31.49 | 26.37 | 0.00 | 2021-02-18 |
| Spain | 178.69 | 84.85 | 81.01 | 29.40 | 2021-01-04 |
| Sweden | 172.96 | 76.14 | 72.68 | 0.00 | 2021-01-03 |
| Switzerland | 158.90 | 68.56 | 66.88 | 24.99 | 2020-12-21 |
| Turkey | 154.80 | 66.92 | 60.68 | 27.19 | 2021-01-14 |
| United Kingdom | 195.45 | 75.93 | 69.54 | 49.98 | 2021-01-10 |
| Vietnam | 153.75 | 79.00 | 69.71 | 0.00 | 2021-03-08 |

Table 8: Information about vaccination on December 31st, 2021, in the countries with presence of Omicron (as defined in Section 2.4.2).

| Country | Total Oct | Total Dec | Unvac Oct | Unvac Dec | Vac Oct | Vac Dec | Vac 1D Oct | Vac 1D Dec | Vac 2D Oct | Vac 2D Dec |
|---|---|---|---|---|---|---|---|---|---|---|
| Argentina | 44509 | 48807 | 3077 | 2778 | 40276 | 44590 | 3704 | 1884 | 36115 | 41783 |
| Belgium | 16448 | 18373 | 1687 | 1718 | 14266 | 16004 | 747 | 463 | 13327 | 15269 |
| Brazil | 198423 | 162402 | 9428 | 6552 | 183859 | 151114 | 38885 | 8680 | 142594 | 139517 |
| Colombia | 34859 | 33883 | 5437 | 2734 | 28457 | 30197 | 9979 | 7514 | 18034 | 22137 |
| Denmark | 19591 | 27284 | 917 | 1206 | 18279 | 25472 | 212 | 217 | 17781 | 24684 |
| France | 82767 | 111041 | 10234 | 11593 | 67393 | 95663 | 6369 | 4708 | 60218 | 89139 |
| Germany | 89348 | 110359 | 12601 | 11868 | 71980 | 95530 | 6655 | 5490 | 64611 | 88548 |
| India | 76675 | 68155 | 4076 | 2631 | 63803 | 60076 | 16798 | 7344 | 45967 | 51622 |
| Italy | 98712 | 112754 | 7023 | 6095 | 89120 | 103305 | 9066 | 5108 | 78852 | 96124 |
| Mexico | 139967 | 118861 | 12063 | 6472 | 119471 | 109330 | 35960 | 17776 | 82321 | 90162 |
| Netherlands | 27505 | 30803 | 3804 | 3380 | 23001 | 26621 | 2175 | 2025 | 20397 | 24087 |
| Norway | 16746 | 21862 | 935 | 1010 | 15536 | 20404 | 389 | 304 | 14980 | 19724 |
| Poland | 30295 | 38001 | 5318 | 6105 | 23924 | 30578 | 2327 | 2499 | 21236 | 27603 |
| Portugal | 22758 | 29352 | 1299 | 1368 | 21017 | 27340 | 3470 | 3172 | 17180 | 23631 |
| Romania | 45123 | 24638 | 11038 | 4917 | 32558 | 19022 | 4477 | 2451 | 27594 | 16192 |
| Russia | 35186 | 30037 | 12301 | 9001 | 21680 | 19884 | 2845 | 2819 | 18573 | 16779 |
| Slovakia | 9567 | 11323 | 1987 | 2208 | 7382 | 8841 | 306 | 487 | 6989 | 8215 |
| South Africa | 18308 | 19492 | 4149 | 4006 | 12805 | 14753 | 5009 | 4138 | 7624 | 10423 |
| Spain | 33455 | 51568 | 2035 | 2625 | 30652 | 47444 | 3814 | 3574 | 26453 | 43223 |
| Sweden | 53564 | 57823 | 3001 | 3200 | 49564 | 53544 | 699 | 443 | 48380 | 52348 |
| Switzerland | 14863 | 16755 | 2906 | 2617 | 11585 | 13742 | 886 | 676 | 10541 | 12824 |
| Turkey | 27159 | 22854 | 3238 | 2307 | 23033 | 19844 | 1473 | 729 | 21015 | 18561 |
| United Kingdom | 41812 | 47072 | 3080 | 3174 | 37421 | 42421 | 925 | 770 | 36109 | 41122 |
| Vietnam | 48955 | 39105 | 8043 | 1116 | 37073 | 36097 | 17325 | 3241 | 19233 | 32246 |

Table 9: Number of survey responses used in each period from the countries with presence of Omicron (as defined in Section 2.4.2), for each level of vaccination.

| Country | Pos Oct | Pos Dec | Unvac Oct | Unvac Dec | Vac Oct | Vac Dec | Vac 1D Oct | Vac 1D Dec | Vac 2D Oct | Vac 2D Dec |
|---|---|---|---|---|---|---|---|---|---|---|
| Argentina | 715 | 1302 | 87 | 99 | 594 | 1143 | 102 | 90 | 484 | 1034 |
| Belgium | 364 | 912 | 69 | 130 | 274 | 751 | 25 | 31 | 248 | 713 |
| Brazil | 5111 | 4066 | 405 | 224 | 4486 | 3648 | 1334 | 355 | 3072 | 3194 |
| Colombia | 1013 | 1103 | 285 | 158 | 666 | 897 | 291 | 280 | 364 | 596 |
| Denmark | 232 | 1405 | 24 | 116 | 196 | 1256 | 5 | 16 | 186 | 1228 |
| France | 703 | 3452 | 149 | 596 | 486 | 2733 | 102 | 130 | 377 | 2566 |
| Germany | 619 | 2253 | 155 | 580 | 428 | 1616 | 52 | 149 | 373 | 1453 |
| India | 2899 | 2231 | 186 | 93 | 1629 | 1235 | 623 | 242 | 958 | 939 |
| Italy | 558 | 2610 | 120 | 329 | 394 | 2158 | 67 | 95 | 322 | 2035 |
| Mexico | 6881 | 4747 | 1201 | 485 | 5167 | 4047 | 2287 | 1038 | 2808 | 2956 |
| Netherlands | 487 | 1441 | 95 | 210 | 367 | 1179 | 60 | 106 | 299 | 1046 |
| Norway | 147 | 569 | 15 | 39 | 127 | 516 | 10 | 17 | 116 | 495 |
| Poland | 1039 | 2504 | 298 | 749 | 676 | 1614 | 90 | 173 | 572 | 1416 |
| Portugal | 170 | 821 | 17 | 55 | 142 | 742 | 28 | 98 | 112 | 632 |
| Romania | 2579 | 448 | 1109 | 175 | 1335 | 239 | 158 | 42 | 1158 | 186 |
| Russia | 1550 | 775 | 752 | 318 | 727 | 401 | 79 | 70 | 633 | 323 |
| Slovakia | 276 | 635 | 89 | 216 | 174 | 397 | 14 | 36 | 157 | 360 |
| South Africa | 695 | 2348 | 249 | 599 | 388 | 1672 | 214 | 564 | 167 | 1093 |
| Spain | 468 | 2776 | 65 | 186 | 375 | 2479 | 80 | 177 | 290 | 2277 |
| Sweden | 297 | 1037 | 48 | 103 | 234 | 899 | 8 | 16 | 225 | 878 |
| Switzerland | 170 | 639 | 61 | 175 | 102 | 445 | 10 | 21 | 90 | 418 |
| Turkey | 1479 | 1143 | 288 | 181 | 1125 | 897 | 136 | 57 | 962 | 818 |
| United Kingdom | 1321 | 2168 | 141 | 180 | 1124 | 1926 | 53 | 59 | 1060 | 1851 |
| Vietnam | 364 | 1271 | 58 | 35 | 251 | 1141 | 95 | 76 | 152 | 1043 |

**Table 10: Number of survey responses classified as positive by Random Forest in each period from the countries with presence of Omicron (as defined in Section 2.4.2), for each level of vaccination.**