# OpenReview forum: "Using Survey Data to Estimate the Impact of the Omicron Variant on Vaccine Efficacy against COVID-19 Infection"
_ACM.org/SIGKDD/2022/Workshop/epiDAMIK — KDD 2022 Workshop epiDAMIK Oral_

### Official Review · Reviewer_FHmh · 2022-06-19
**Technically sound, need several clarifications on approaches and results**

**Rating:** 4
**Confidence:** 4

**Review:**

This is an interesting study that used symptom surveys to estimate the vaccine efficacy before and after the Omicron variant emerged. The results qualitatively agree with evidence from recent clinical studies. I have a few questions that need the authors to clarify.

1. The random forest uses symptom information to classify positive and negative. As we know, many infections may have no symptoms. How were asymptomatic infections handled in the model?

2. What’s the fraction of missing values in symptom surveys? How did you deal with missing values?

3. Have the authors considered different vaccines administered in different countries?

4. In Fig. 2b, the VE has very large fluctuations after November. This does not make sense given that the dominant strain was Omicron, and the situation was relatively stable. Could the authors explain the drivers of this fluctuation?

5. I suggest the authors compare the VE estimate with published studies that used clinical/cohort data. It would be good to put the findings into context and see the discrepancy among studies.

---

### Official Review · Reviewer_Pane · 2022-06-27
**This paper uses the self-reported survey data to evaluate the prevalence of the Omicron variant to see the effectiveness of COVID vaccines.**

**Rating:** 4
**Confidence:** 4

**Review:**

This paper uses the self-reported survey data to evaluate the prevalence of the Omicron variant to see the effectiveness of COVID vaccines. Specifically, the author explores the prevalence of Omicron infections among different kinds of vaccinated people (1-dose, 2-dose, vaccinated, and unvaccinated).

Pros:
1. The author noticed that the ground-truth dataset cannot be used directly to estimate the prevalence of COVID-19 in the overall population. Hence, the author proposed to use the random forest to generate more labels (and hence data) for evaluation.
2. The idea of using CLI data to estimate the prevalence of different COVID variants is novel.
3. The experiments are performed in many counties, which makes the conclusion more convincing.

Cons:
1. The experiments are also based on the patients with COVID-related people, so no evaluation of other people (i.e. no symptoms) are evaluated.
2. The comparison of random forest with other classifiers (SVM, neural network...) can be also helpful.